# Modified Serum ALP Values and Timing of Apparition of Knee Epiphyseal Ossification Centers in Preterm Infants with Cholestasis and Risk of Concomitant Metabolic Bone Disease of Prematurity

**DOI:** 10.3390/nu12123854

**Published:** 2020-12-17

**Authors:** Sandra Llorente-Pelayo, Pablo Docio, Bernardo A. Lavín-Gómez, María T. García-Unzueta, Isabel de las Cuevas, Luis de la Rubia, María J. Cabero-Pérez, Domingo González-Lamuño

**Affiliations:** 1Pediatric Department, University Hospital Marqués de Valdecilla—Research Institute Valdecilla (IDIVAL), 39008 Santander, Spain; sandra.llorente@scsalud.es (S.L.-P.); pablo.docio@scsalud.es (P.D.); 2Biochemical Department, University Hospital Marqués de Valdecilla—Research Institute Valdecilla (IDIVAL), 39008 Santander, Spain; bernardoalio.lavin@scsalud.es (B.A.L.-G.); mteresa.garciau@scsalud.es (M.T.G.-U.); 3Neonatology Unit, Pediatric Department, University Hospital Marqués de Valdecilla—Research Institute Valdecilla (IDIVAL), 39008 Santander, Spain; misabeldelas.cuevas@scsalud.es; 4Departmento de Ciencias Médicas y Quirúrgicas, University of Cantabria, 39005 Santander, Spain; luisramondela.rubia@scsalud.es; 5Gastroenterology and Hepatology Unit, Pediatric Department, University Hospital Marqués de Valdecilla—Research Institute Valdecilla (IDIVAL), 39008 Santander, Spain

**Keywords:** metabolic bone disease, prematurity, osteopenia, rickets, cholestasis, alkaline phosphatase, bone specific alkaline phosphatase, epiphyseal ossification centers

## Abstract

The usefulness of serum alkaline phosphatase (ALP) and phosphorous in screening and monitoring of metabolic bone disease of prematurity (MBDP) still has some limitations, especially in preterm infants with concomitant conditions such as cholestasis. We aimed to assess a modification of serum ALP (M-ALP) as a biomarker for MBDP in preterm infants, and the use of ultrasound monitoring for the apparition of knee ossification centers as marker of bone mineralization. Biochemical and clinical registers were taken from 94 preterm newborns <32 weeks. A significant correlation existed between serum ALP and direct bilirubin (DB), expressed by the regression equation: M-ALP (IU/L) = 302.1 + 96.9 (DB (mg/dL)). The ratio ALP/M-ALP > 1 was demonstrated to be more specific (87.5%) in the diagnosis of MBDP than the cut-off value of serum ALP > 500 IU/L (62.5%). ALP/M-ALP > 1 showed 100% sensitivity and specificity for the diagnosis of MBDP, and a good correlation with specific bone ALP (B-ALP). Patients with the knee nucleus by post-menstrual week 37 had lower B-ALP compared to patients with no nucleus, and no patients with MBDP presented the nucleus by the 40th week. In the absence of reliable specific B-ALP, reinterpreting serum ALP values by M-ALP plus monitoring of knee ossification centers contribute to better management of MBDP in preterm infants with cholestasis.

## 1. Introduction

Metabolic bone disease of prematurity (MBDP), also known as osteopenia or rickets of prematurity, is a condition characterized by reduction in bone mineral content [1,2]. Because of the lack of fetal mineralization during the last trimester, it is frequently observed in preterm infants born before completing 32 weeks’ gestation [3], occurring in 16–60% of very low birth weight (VLBW, <1500 g) and extremely low birth weight (ELBW, <1000 g) infants [4,5,6].

There is no consensus with regard to optimal screening, diagnosis, and timing of treatment initiation in MBDP [6,7,8]. However, neonatologists recognize the importance of maximizing bone health in preterm infants, thus fortified breast milk and formulas designed for premature infants are generally used to guarantee adequate calcium and phosphorus intake by a preterm infant [8]. The clinical signs of MBDP appear between 6–16 weeks after birth [5], but the assessment of serum biochemical markers is useful for early detection of mineral deficiency in the first 4–6 weeks of life. A significant loss of bone mineralization is needed before characteristic changes are visible on radiography [9].

Choosing whom to screen for MBDP is driven by risk factors (birth weight (BW) <1500 g; gestational age (GA) <28 weeks; parenteral nutrition >4 weeks; unable to reach full fortified feeds; or use of bone active medications), but deciding the best screening test (serum calcium, phosphorous, alkaline phosphatase (ALP), tubular reabsorption of phosphate (TRP)) can be a challenge [8,10,11].

Biochemical characteristics of high serum ALP and low serum phosphate are compatible with MBDP. Whereas in healthy individuals total serum ALP is derived predominately from the liver and bones (more than 80%), ALP derived from bones (B-ALP) is a reliable biomarker of bone turnover and predicts the status of bone mineralization. High levels of B-ALP entail increased bone cellular activity and are linked to osteopenia, which may not yet be clinically apparent, but indicates the need for further evaluation and individualized mineral supplementation in premature infants, particularly those of <1000 g birth weight and <32 weeks’ gestation [12,13].

As determinations of B-ALP are not included in most laboratories’ routines, biochemical screening for osteopenia of prematurity is based on total serum ALP alone, which is suggestive of rickets at very high levels of >1000 IU/L [14]. At lower levels, specificity for MBDP improves when serum ALP is used in combination with serum phosphorus levels (<5.6 mg/dL or <1.8 mmol/L). Some studies have denoted that serum ALP > 500 IU/L and trending up has a variable diagnostic accuracy for detecting osteopenia, with a sensitivity of 100% and a specificity of 70–80% [15,16] and, although not universally accepted, it is the most used cut-off value of ALP for osteopenia. In cases of simultaneous presence of bone and liver disorders, is necessary to measure the activity of the different isoenzymes (mainly B-ALP and liver ALP (L-ALP)), indicating the rate and the extent of the relative tissue damage.

The evaluation of skeletal mineralization could also help in the assessment of the timing and intensity of mineral supplementation. Although using a non-ionizing method is desirable, presently, dual energy x-ray absorptiometry (DXA) is the gold standard method for the assessment of bone mineralization in newborns [17,18]. Radiographs (Rx) can reveal various degrees of MBDP, including demineralization or osteopenia, rachitic changes, and/or fractures [8,19]. Measurements of the ossification center in the distal femoral (DFE) and proximal tibial epiphysis (PTE) by ultrasound is an excellent tool for assessing skeletal maturation in premature newborns. The proportion of fetuses in which the DFE is detectable by ultrasonography is 71–72% at 32–33 weeks, reaching 91–94% at 34–35 weeks and 100% at 37 weeks gestation. Up to 36 weeks’ gestation, the PTE is detectable in only 33–50% of fetuses, but at 37 weeks it is observed in more than 80%, increasing to 97% at 39 weeks and 100% at 40 weeks’ gestation [20,21]. According these data, the absence of DFE in a preterm infant with a post-menstrual age of 34–36 weeks or absence of PTE at 38–39 weeks indicates a high suspicion of MBDP.

Although there are no published data, in preterm infants <32 weeks’ with MBDP, ossification is almost invariably delayed until appropriate mineralization exists. A detailed morphometric follow-up on the PTE and DFE nucleus by a non-invasive and consistent method would be useful for the monitoring and timing of mineralization in infants with MBDP under treatment.

Neonatal cholestasis, manifested by an increased conjugated bilirubin fraction, refers to the accumulation of bilirubin and bile acids as a result of impaired bile flow that occurs in up to 18–24% of VLBW infants [22]. In preterm infants, neonatal cholestasis has different and multifactorial etiology as compared with term infants, being most often related to feeding intolerance and prolonged use of parenteral nutrition [23]. According to the best of our knowledge, there are no consistent reports on the specificity of serum ALP values in preterm infants with concomitant MBDP and cholestasis. We aimed to assess a modification of serum ALP (M-ALP) values as a potential marker for MBDP in preterm infants with concomitant cholestasis. In preterm infants with cholestasis, an ALP value modified by serum DB should modify the specificity of serum ALP for MBDP, thus favoring optimization in the timing and intensity of mineral supplementation. Although there are no normalized values for B-ALP in preterm infants, and its determination is not accessible in all centers, we here investigate the specificity of this parameter compared with the M-ALP in a small number of samples from preterm infants with and without biochemical criteria of MBDP. Finally, we propose the use of ultrasound monitoring for the apparition DFE and PTE ossification centers as a new marker of adequate bone mineralization, which should be valuable in deciding the timing and intensity of MBDP treatments.

## 2. Patients and Methods

### 2.1. Patients

The present study was conducted on different historical series of preterm infants of both sexes, born and hospitalized for at least 4 weeks in the neonatal unit of the University Hospital Marqués de Valdecilla, in Santander, Spain. Registers were obtained from different cohorts of 94 preterm infants. These included a cohort of 14 preterm newborns with a diagnosis of cholestasis (cholestasis group, ChG) in the period of 2012 to 2015; registers from 70 preterm newborns from 2016 to 2019 included in the named retrospective group (RG); and a prospective group (PG) of 10 preterm newborns recruited in 2020. The current study included serum and urine analysis from the first two months of life of all preterm newborn infants with GA < 32 weeks and/or BW < 1500 g. Patients with skeletal deformities were excluded. Samples with incomplete hepatic profiles were not considered.

The scientific research committee of the University Hospital-Instituto de Investigación Valdecilla (IDIVAL) approved the study. When appropriate (prospective study-group, PG), the consent included an agreement for assessment of knee ultrasonography performance.

### 2.2. Methods

The biochemical criteria established for MBDP was a serum phosphorus level lower than 5.6 mg/dL (or 1.8 mMol/L), plus total serum ALP above 500 IU/L. Criteria for cholestasis was defined as serum direct bilirubin >2 mg/dL or >20% of the total serum bilirubin level when the total was elevated.

Medical records were examined retrospectively in all enrolled infants between 2012 and 2019, and registered prospectively in those enrolled in 2020, including full medical history, anthropometric measurements, and assessment of GA. Complications such as neonatal cholestasis, sepsis, bronchopulmonary dysplasia, necrotizing enterocolitis (NEC), or gastrointestinal perforation were recorded. Biochemical data determined (ADVIA2400 platform, Siemens Healthcare, Malvern, PA, USA) and recorded included bone profiles with serum calcium, phosphorus, ALP and vitamin D (calcidiol) and calcinuria, phosphaturia and TRP; liver profile with alanine transaminase (ALT), aspartate transaminase (AST), direct and total bilirubin (DB and TB), gamma glutamyl transferase (GGT), and albumin. In the PG study group, bone biochemical profiles were expanded to parathyroid hormone (PTH), B-ALP, fibroblast growth factor-23 (FGF23), 25-hydroxycholecalciferol (calcidiol) and 1,25-dihydroxycholecalciferol (1,25(OH) 2-vitamin D3 or calcitriol), functional renal tubular determinations, and a periodic (weekly) ultrasonographic knee monitorization investigating the presence of both the PTE and DFE nucleus.

Ultrasonographic assessment was performed using a wide-band linear transducer with ultrasound scanner Philips Affiniti 50, by two different general pediatricians separately. There were no intra- or inter-observer differences in the results. Before the study, both pediatricians had been trained in the US nucleus assessment by performing echography on healthy full-term newborns. Each patient was measured twice by both operators (general pediatricians) on two successive days. Presence of an epiphyseal ossification center was defined by the presence of a measurable hyper-echoic structure with an irregular or smooth surface within the epiphyseal cartilage. When present, longitudinal and transverse images of the left knee centers were obtained with the patient in supine position and the limb extended.

A correlation analysis between ALP and DB using a linear regression model was performed in the RG, in order to establish an M-ALP value for predicting MBDP in patients with concomitant cholestasis. The correlation model was tested in preterm infants from the ChG and PG groups, comparing the sensibility and specificity for the diagnosis of MBDP with the classical cut off point of ALP > 500 IU/L and the corrected ALP value calculated with the model.

Data analysis was performed using SPSS 17.0.1 SPSS Inc, Chicago, USA. Data with a normal distribution were expressed as the mean ± SD; data with a non-normal distribution were expressed as the median ± IQR. In the comparative analysis, inter-group comparison was performed using Student’s t in normal cases, and the U by Mann Whitney test in non-normal cases. Furthermore, a correlation and lineal regression model was used. *p* < 0.05 was considered to indicate statistical significance.

Informed consent was obtained from all individual participants (from their parents or guardians) included in the study according to the criteria of the Ley de Investigación Biomédica 14/2007 de 3 de Julio. The study was approved by the Ethical Committee of the authors’ hospital (CEIC-C: Comité Ético de Investigación Clínica de Cantabria, project number CSI20/09)

## 3. Results

The RG registers from 2016 to 2019 included 115 different biochemical studies from 70 preterm patients of less than 32 weeks. The correlation between serum ALP and DB values was analyzed in a linear regression model, excluding both results from samples compatible with MBDP (*n* = 18) and incomplete samples with an absence of DB values. A final number of 77 samples corresponding to 49 different infants were used for the correlation analysis. A total of 50 samples (64.9%) were taken in the first month of life, and 27 samples (35.1%) were taken in the second month of life.

Of the 49 patients included, 26 (53.1%) were female. The median GA of the patients was 30 weeks, ranging from 26 to 31 weeks, with a median weight at birth of 1220 g, ranging from 730 to 1805 g. A total of 18.4% of the patients (9/49) presented different grades of neonatal cholestasis. Other frequent diagnoses were apnea of prematurity in 42 patients (85.7%); hyaline membrane disease in 36 patients (73.5%); bronchopulmonary dysplasia in 11 patients (22.4%); jaundice of prematurity in 24 patients (49%); anemia of prematurity in 21 patients (42.9%); hypoglycemia in 20 patients (40.8%); neonatal sepsis in 13 patients (26.5%); permeable oval foramen in 12 patients (24.5%); patent ductus arteriosus in 8 patients (16.3%); and hypothyroidism in 8 patients (16.3%).

As expected, in the RG we found a statistically significant difference in serum ALP values between samples of patients diagnosed with cholestasis and those without cholestasis, with no differences in terms of phosphate, calcium, urinary phosphate, or phosphaturia (Table 1). We also found a linear correlation between the levels of ALP and DB (Figure 1) expressed by the regression equation: M-ALP (IU/L) = 302.1 + 96.9 [DB (mg/dL)], with a correlation coefficient of 0.682 (*p* < 0.001).

Retrospectively, among the group of 14 preterm patients with cholestasis (ChG) we calculated the M-ALP using the previous equation (Table 2). Maintaining sensitivity for the diagnosis of MBDP of 100%, the ratio ALP/M-ALP > 1 was demonstrated to be more specific (87.5%) for the diagnosis of MBDP in preterm infants with concomitant cholestasis than the cut-off value of serum ALP > 500 IU/L (62.5%) (Table 3).

For the diagnosis of MBDP in patients with cholestasis, the cut-off value of serum ALP > 500 IU/L (a) has 100% (6/6) sensitivity but 62.5% (5/8) specificity; while the ratio ALP/M-ALP > 1 (b) has 100% (6/6) sensitivity with a specificity of 87.5% (7/8).

In the premature PG group, serum ALP levels at a biological age of 4 weeks ranged from 230 to 745 IU/L (median 380 IU/L), with B-ALP levels that ranged from 87 IU/L to 354 IU/L (median 163 IU/L), supposing a percentage of the total serum ALP ranging from 36.9% to 47.5%. Three of the patients presented with ALP > 500 IU/L. In the only patient with cholestasis (Table 2, patient number 2), the ratio of ALP/M-ALP > 1 had good sensitivity and specificity for the diagnosis of MBDP and showed a good correlation with the B-ALP. As expected, the percentage of B-ALP over the serum ALP (36.9%) was lower in the patient with cholestasis. Regarding the presence of ossification nucleus in the same PG group, only 3 patients (30%) presented the DFE nucleus by the 37th post-menstrual week, and only 4 presented the DFE + PTE nucleus by the 40th post-menstrual week. Concordance between both observers in ultrasound results was 100% in assessing the visibility of the nucleus. None of the patients diagnosed with MBDP with the classical criteria during the first two months of life presented the nucleus by the 40th post-menstrual week. We observed that patients with the DFE nucleus by week 37 showed lower serum levels of B-ALP at week 3–4 compared to patients with no nucleus (Table 4 and Figure 2).

### 3.1. Clinical Case Examples (A, B and C) from the Prospective Group (PG)

In clinical practice, when managing a preterm infant with potentially both MBDP and cholestasis, due to the dynamism of the first weeks of life there can be difficulties in the interpretation of biochemical routine analysis. As a clinical example (case A), in our PG assessment, a 26 4/7 GE female’s analytics, at 4 weeks of life, showed serum phosphorus 5.2 mg/dL, serum calcium 9.6 mg/dL, ALP 230 IU/L, B-ALP 88 IU/L, TB 0.9 mg/dL (DB 0.5 mg/dL), urine phosphorus 9 mg, and TRP 90%. At that point, criteria for neither MBDP nor cholestasis were fulfilled. Forty nine days later (post-menstrual age, PMA, 36 6/7), after suffering some conditions which contribute to cholestasis, a second-time sample was recorded showing the following: phosphorus 4.8 mg/dL, calcium 9.4 mg/dL, ALP 416 IU/L, TB 1.3 mg/dL, DB 0.9 mg/dL, urine phosphorus <4 mg, and TRP at least 98.3%; when implementing our formula, M-ALP was 389.3 IU/L with a ratio of ALP/M-ALP > 1. A diagnosis of cholestasis was established, whereas MBDP was not considered based on serum ALP. Considering the ALP/M-ALP ratio of >1, the infant had concomitant MBDP, and ultrasonography showed there was absence of both ossification nuclei.

It is not only important to screen for MBDP, but also to carry out suitable monitoring once the diagnosis is confirmed and when setting up treatment. As is widely acknowledged, a suitable intake of calcium and phosphorus is recommended in order to prevent imbalances that can lead to NC.

As an example (case B), we evaluated the records of a 28 5/7 GE female in order to interpret calculated M-ALP. Initial values of serum phosphorus were 4.6 mg/dL, serum calcium 8.9 mg/dL, ALP 601 IU/L, B-ALP 222 IU/L, TB 3.6 mg/dL, and DB 2.8 mg/dL; M-ALP 573.4 IU/L and the ratio of ALP/M-ALP was >1. Urine phosphorus was 14 mg, with a TRP of 73%. Given these biochemical data, phosphorus supplementation was initiated, and a wait-and-see approach was adopted with liver parameters monitoring. Twenty-one days later, ursodeoxycolic acid was begun, as liver function tests indicated cholestasis (TB 5.2 mg/dL, DB 4.4 mg/dL, ALP 351 IU/L and M-ALP 728.46 IU/L, thus ALP/M-ALP < 1). In subsequent analysis we noticed a stabilization of the ALP values, ranging from 380 to 480 IU/L, and a gradual decrease in DB values. Phosphorus values ranged from 3.3 to 5.7 mg/dL, with TRP ranging from 94% to 99.2%. Ultrasound monitoring was used to assess the presence of the PTE and DFE nucleus at PMA 44 3/7. As bone maturation is completed, mineral and vitamin D supplementation should be re-adjusted (diminished) in order to avoid over-treatment, which could lead to NC.

Focusing on the appearance of NC in potentially over-supplemented infants, when assessing medical records retrospectively we found (case C) a 26 3/7 GE male whose first sample showed serum phosphorus 3.5 mg/dL, serum calcium 10 mg/dL, ALP 1320 IU/L, TB 7.1 mg/dL, DB 5.4 mg/dL, and urine phosphorus <4 mg. When using the equation, M-ALP was 825.36 IU/L and the ratio of ALP/M-ALP was >1. Phosphorus supplementation, as well as supplementation of vitamins A, E and K plus ursodeoxycolic acid, were initiated. Several analyses performed at 50, 75 and 87 days of life showed a remarkable elevation of both phosphorus (6.1, 5.8, and 6.2 mg/dL, respectively) and ALP values (693, 641, and 522 IU/L, respectively). Urine phosphorus was significantly higher (29, 37, and 48 mg, respectively). The calculated M-ALP values were 835.05, 738.15 and 573.42 IU/L, with all ratios <1, consistent with improvement of the MBDP and cholestasis. However, phosphorus supplementation was continued at a treatment dose regimen, and at 87 days of life, bilateral NC was detected by abdominal ultrasonography.

### 3.2. Formatting of Mathematical Components

M-ALP (IU/L) = 302.1 + 96.9 (DB (mg/dL))
Regression equation of the linear correlation between the levels of ALP and DB with a correlation coefficient of 0.682 (*p* < 0.001).(1)

## 4. Discussion

In this study, we proposed a modified interpretation of ALP values in order to increase the specificity of ALP values in the diagnosis of MBDP in preterm infants with cholestasis. In a prospective analysis of a historical cohort of preterm infants with cholestasis (ChG), the ratio of ALP/M-ALP of >1 was significantly more specific (87.5%) in the biochemical diagnosis of MBDP than a cut-off value of serum ALP > 500 IU/L (62.5%). In a prospective pilot study of preterm infants, M-ALP correlated with the values of B-ALP. We also proposed the use of ultrasonography of knees for the timing of presentation of the ossification nucleus of the PTE and DFE as new objective markers for adequate bone mineralization. The description of three representative examples from the study cohort are included in the manuscript to show the potential clinical relevance of our study in preterm infants with cholestasis, and at risk of MBDP.

In preterm infants, different grades of both cholestasis and osteopenia are dynamic conditions that occur in the first weeks of life, so the adequate interpretation of serum ALP continues to be a challenge. Although hyper-phosphatasia in the first weeks of life typically represents inadequate mineral intake and often accompanies MBDP, serum ALP levels have not been shown to correlate to the degree of hypo-mineralization. Rachitic changes are more commonly associated with ALP levels of more than 800 IU/L but can also be seen at lower levels of less than 600 IU/L [14]. Preterm infants with prolonged jaundice and cholestasis are also at risk of MBDP, and the need for additional minerals and vitamin D supplements may not be apparent or addressed, thus illustrating the importance of improving the specificity of ALP determinations.

As cholestasis is a relatively common concomitant condition in preterm infants at risk of MBDP, in order to discriminate both conditions it is necessary to obtain serial specific B-ALP measurements during the first two months of life. Since there are no normalized values for B-ALP and it is not plausible to obtain this biomarker in most hospitals, our calculated ALP/M-ALP ratio corrects serum ALP for the expected increase in L-ALP. As expected, a significant correlation between DB and serum ALP in the cohort of preterm infants with adequate grades of bone mineralization was found, expressed by the following regression equation: M-ALP (IU/L) = 302.1 + 96.9 (DB (mg/dL)), with a correlation coefficient of 0.682 (*p* < 0.001).

The goal for prevention of MBDP is to maintain the bone accretion rates, biochemically monitored by serum levels for both calcium and phosphorus that should be maintained in normal levels, but also reflected in serum ALP levels in the normal range for the age. In MBDP elevated serum, ALP exhibits 100% sensibility but a variable degree of specificity in the context of cholestasis or other associated conditions commonly presented as co-morbidities of preterm infants. Thus, in preterm infants with a risk of MBDP, once mineral supplementation or other treatment is started, monitoring and therapeutic goals should be individualized to the underlying deficiency and specific treatment. Although there are no conclusive monitoring guidelines, it seems reasonable to monitor serum phosphorus and ALP periodically (weekly or biweekly) [8]. Mineral supplementation by using fortified breast milk and preterm formulas are generally preferred by neonatologists, as formulas not designed for premature infants do not have an adequate calcium and phosphorus content [8].

Clinical follow-up and monitoring of biochemical markers are mandatory both for diagnosis and assessment of individual therapeutic responses, and to screen for complications of mineral supplementation. It is important to note that electrolyte imbalances and rapid augmentation of calcium and phosphorus absorption could occur when mineral supplies are increased too quickly, or after prolonged periods of interrupted enteral nutrition [24]. Hypophosphatemia and hyper-phosphatasia are the most common biochemical changes of MBDP, and are biomarkers indicating insufficient mineralization. As early as 7–14 days after birth, newborns can exhibit hypophosphatemia as an early marker of disrupted mineral metabolism [25]. Phosphate deficiency suppresses parathormone (PTH), thereby preventing urinary phosphate wasting but activating synthesis of 1,25(OH) 2-vitamin D3 that leads to an increase in intestinal calcium and phosphate reabsorption. Thus, phosphate deficiency disrupts calcium balance, potentially leading to hypercalcemia, hypercalciuria, and nephro-calcinosis (NC).

The prevalence of NC in preterm neonates with GA < 32 weeks or BW < 1500 g is highly variable, from 7% to 64% [26,27,28,29], due not only to differences in the study populations and ultrasound criteria, but also to a moderate inter-observer variation. NC in preterm neonates has a multifactorial etiology, consisting of low GA and BW, often in combination with severe respiratory disease, and occurs as a result of an imbalance between stone-promoting and stone-inhibiting factors. Stone-promoting factors such as hypercalciuria and hyperoxaluria, as described below, in combination with reduced stone-inhibiting factors like low urinary citrate excretion, can lead to NC.

The recommended phosphorus intake for preterm neonates is 60–90 mg/kg per day and, like calcium intake, is high in comparison with that for term neonates [30]. For infants with hypophosphatemia, phosphorus supplementation can be adjusted to reach a target serum phosphorus of >5.5 mg/dL.

Low intake of phosphorus can result in hypophosphatemia, leading to a rise in phosphorus reabsorption in the proximal tubules, an increase in calcium and phosphorus absorption from the bone, and an increase in renal production of 1,25(OH) 2-vitamin D3. Vitamin D stimulates the intestinal absorption and resorption of calcium and phosphorus from the bone. The consequent increase in serum calcium induces suppression of PTH release, which results in a further decrease in urine phosphate and an increase in calcium excretion in adults [31]. In contrast, a high intake of phosphorus is also associated with NC in preterm neonates [21,32]. In patients with hypo-phosphatemic rickets, high intake of phosphate has been described as leading to hyperoxaluria and NC, even in a normocalciuric state [33].

Isolated plasma calcium levels may not be a helpful screening marker for infants at risk for MBDP, as a normal plasma level does not ensure adequate mineral intake. In order to prevent MBDP and according to the current guidelines, the recommended daily intake of vitamin D in preterm neonates is relatively high (800–1000 IU) [8,30]. Although, vitamin D alone has not been found to be a significant risk factor for NC [31], vitamin D excess can result in hypercalcemia and hypercalciuria. Moreover, in preterm infants, metabolic changes associated with secondary hyperparathyroidism led to urinary phosphate wasting, and an associated low TRP with hypophosphatemia. Increased PTH levels will trigger increased plasma calcium levels due to increased bone resorption and increases in renal and intestinal calcium absorption.

Special considerations are high-risk preterm infants (i.e., infants who do not tolerate enteral feeding and require prolonged total parenteral nutrition, treatment with corticosteroids, methylxanthines and diuretics) with no initial biochemical markers for MBDP, but with cholestasis or prolonged jaundice and elevated serum ALP, attributed to the cholestasis, with retarded ossification [34,35]. A complete metabolic bone profile should be performed and the mineral and vitamin supplements should be optimized.

As MBDP advances, biochemical changes intensify. These changes commonly include hypophosphatemia, hyper-phosphatasia, and secondary hyperparathyroidism, which may be accompanied by rachitic changes and/or fractures. However, it may go unrecognized, as significant loss of bone mineralization is needed before characteristic changes will be visible on radiography [9]. Inadequate bone mineralization during this period may compromise pulmonary status and contribute to poor growth.

Radiography, typically done as part of standard clinical care, can reveal various degrees of MBDP, including demineralization or osteopenia, rachitic changes, and/or fractures. While some fractures can be acute with associated pain or diminished movement, fractures are more commonly healing and without associated symptoms [19]. The American Academy of Pediatrics (AAP) Clinical Report recommends rechecking radiographs every 5–6 weeks until improved mineralization is achieved [8].

The evidence supports the relevance of having a sensible, specific, and reliable protocol to screen and diagnose these conditions, but also to optimize the type of diet and the dose and timing of mineral and vitamin supplements. For infants on phosphorous, calcium, vitamin D, or calcitriol treatments, the goals are to normalize PTH levels, monitor for hypercalciuria, and to normalize phosphorus levels by limiting urinary phosphorus wasting, as evidenced by increases in TRP. High levels of serum ALP and the ALP/M-ALP ratio can be considered a reliable biomarker to predict the status of bone mineralization and the need for radiological evaluation in premature infants, particularly those with a <1000 g birth weight and at <32 weeks’ gestation. However, with ultrasonography we are able to define the grade of bone maturation and mineralization. The evaluation of DFE and PTE nucleus by ultrasonography is harmless, easily reproducible and does not require a high level of expertise by performers but does provide valuable information. As in other studies [36], inter- and intra-observer agreement for visibility of the knee ossification centers was 100%. When mineralization is adequate at a post-menstrual age of 40 weeks, the ossification center is easily identified in all infants, whereas it is retarded in MBDP.

The present study has some limitations as it was conducted in a cohort of 94 preterm infants hospitalized in a University Hospital for more than eight weeks but from three different consecutive periods: 2012–2015, 2016–2019 and 2020. Although there were no changes in the clinical criteria, biochemical protocol determinations or laboratories ranges, each period comprised different preterm infants thus are not strictly comparable. In the cholestatic group (ChG), preterm infants with this specific condition were included during the hospitalization period and data were strictly used to test sensitivity and specificity of the ALP/M-ALP ratio. Biochemical and clinical data from preterm infants in the retrospective group (RG) were used to obtain the M-ALP formula based in ALP and DB correlation. The prospective group (PG) in which knee epiphyseal ossification was tested only included 14 preterm infants. Other medical conditions such as hypoglycemia or hypothyroidism, as being adequately corrected during hospitalization, are not considered in our study.

In the absence of reliable specific B-ALP values, reconsidering serum ALP values in preterm infants with cholestasis, plus sequential monitoring of the knee ossification centers, will contribute to better management of MBDP. In the future, prospective studies in preterm infants will include other variables such as 1,25(OH) 2-vitamin D3, FGF23, and reliable biomarkers of tubular maturation. FGF23 is a phosphaturic agent from osteoblasts, signaling bone status.

## 5. Conclusions

The inadequacy in the first weeks after birth of both radiography and current biochemical markers are medical challenges in the precious detection and optimal treatment of MBDP in preterm infants with concomitant cholestasis. 

As serum bone alkaline phosphatase (B-ALP) is not a routine biochemical determination, we propose the use of a modified value of serum alkaline phosphatase (M-ALP), being M-ALP (IU/L) = 302.1 + 96.9 (DB (mg/dL)), for the screening of MBDP in preterm infants with cholestasis. In these infants, the ratio ALP/M-ALP >1 has 100% sensitivity and is more specific than ALP values (87.5% vs. 62.5%) for the diagnosis of MBDP.

The evaluation of knee ossification centers by ultrasonography is harmless, easily reproducible and does not require a high level of expertise. When mineralization is adequate at a post-menstrual age of 40 weeks, the ossification centers are easily identified in all infants, whereas it is retarded in MBDP.

The dual approach combining measures of M-ALP with ultrasound monitoring of ossification centers favors the optimization (timing and intensity) of MBDP treatments.

## Figures and Tables

**Figure 1 nutrients-12-03854-f001:**
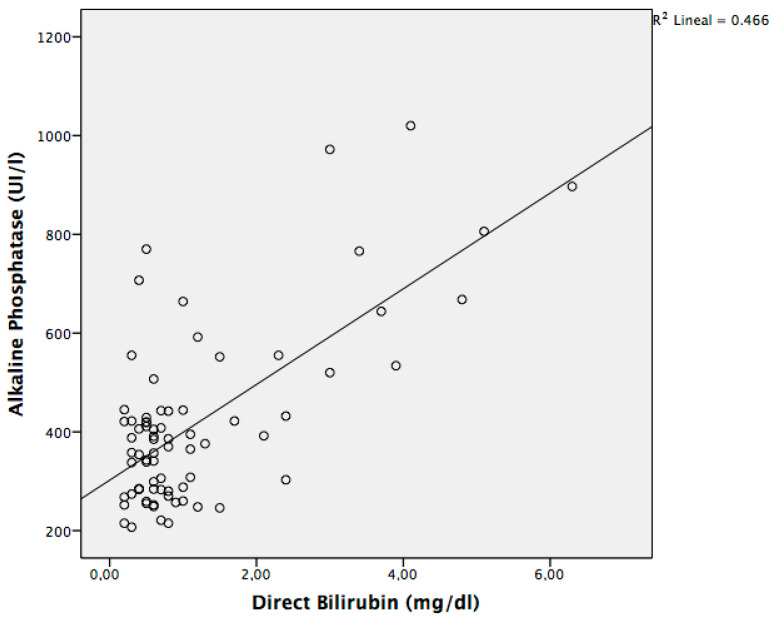
Dispersion graph with the relationship between serum alkaline phosphatase (ALP) and direct bilirubin (DB). Changes in the levels of DB are associated with changes in serum ALP values, with a R^2^ coefficient of determination of 0.466.

**Figure 2 nutrients-12-03854-f002:**
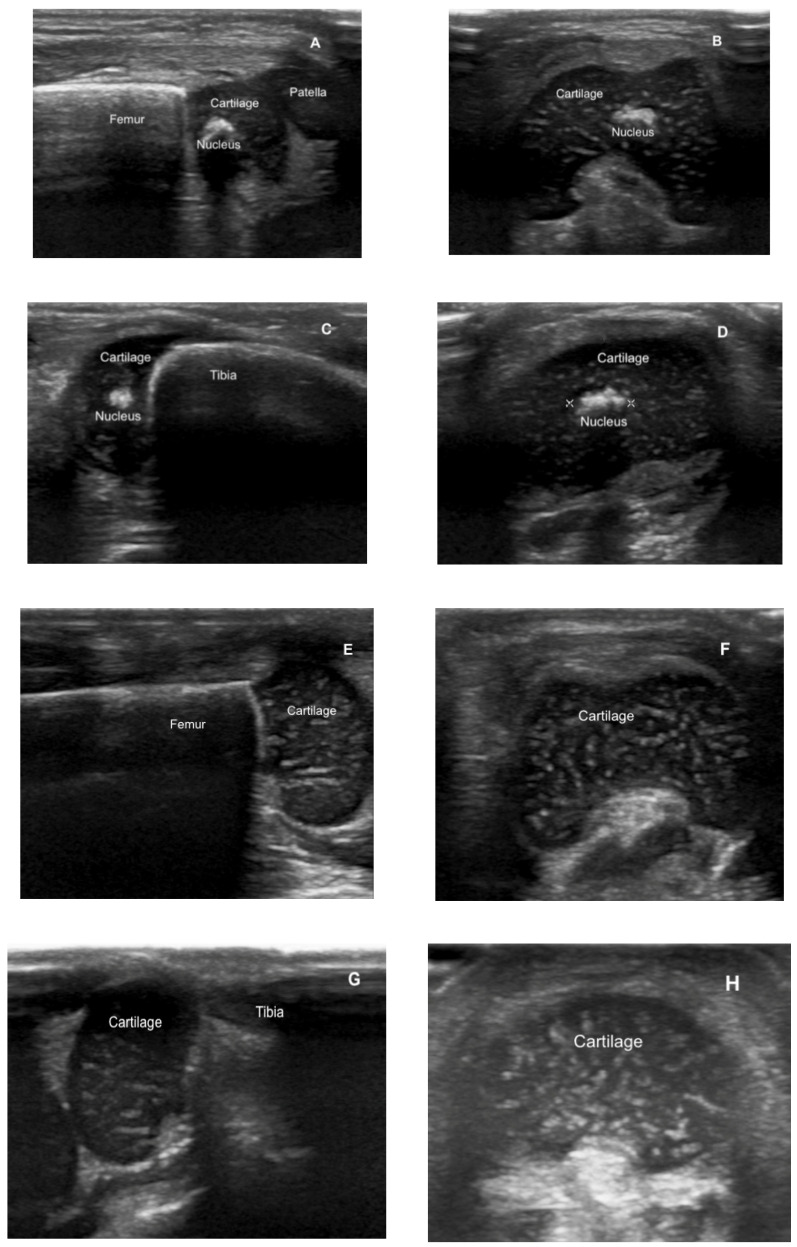
Eco-graphic views of the distal femoral epiphysis (DFE) and the proximal tibial epiphysis (PTE). Pictures A-D show the presence of the DFE nucleus in longitudinal (**A**) and transverse (**B**) views and the PTE nucleus in longitudinal (**C**) and transverse (**D**) views and correspond to a Caucasian 31 + 2 GE male, at 36 postmenstrual weeks. Pictures (**E**–**H**) show the DFE cartilage without nucleus in longitudinal (**E**) and transverse (**F**) views and the PTE cartilage without nucleus in longitudinal (**G**) and transverse (**H**) views.

**Table 1 nutrients-12-03854-t001:** Biochemical determinations from preterm infants in the Retrospective Group (RG). Comparison of the different biochemical values according to cholestasis and non-cholestasis criteria.

	Cholestasis (*n* = 26)	Non Cholestasis (*n* = 51)	Total (*n* = 77)	*p* Value
Alkaline Phosphatase (IU/L)	482 (350.75–665)	343 (274–419)	385 (284–444.5)	0.001 ^1^
Phosphate (mg/dL)	6.27 ± 0.72	6.59 ± 0.68	6.48 ± 0.71	0.061 ^2^
Albumin adjusted Calcium (mg/dL)	10.4 (10–10.6)	10.5 (10.2–10.6)	10.4 (10.2–10.6)	0.204 ^1^
Urinary Phosphate (mg/dL)	11.06 ± 11.39	11.21 ± 10.82	11.17 ± 10.85	0.966 ^2^
P/Cr (mg/mg)	0.99 ± 0.77	1.27 ± 0.92	1.19 ± 0.89	0.359 ^2^
Urinary Calcium (mg/dL)	4.15 (3.08–6.2)	5.4 (3.35–16.6)	4.9 (3.4–14.4)	0.279 ^1^
Ca/Cr (mg/mg)	0 (0–0.5)	0.25 (0–0.7)	0.22 (0–0.62)	0.192 ^1^

^1^ U by Mann Whitney; ^2^ Student’s T: Data are shown as the median (IQR) or mean ± SD according to variable distribution. Abbreviations: P: Phosphate. Ca: Calcium. Cr: Creatinine. SD: Standard deviation. IQR: Interquartile range.

**Table 2 nutrients-12-03854-t002:** Preterm features from the Cholestasis Group (ChG) with alkaline phosphatase (ALP) and modified ALP (M-ALP) values, and the condition of Metabolic Bone Disease of Prematurity (MBDP).

Patient	Gender	GA (Weeks)	BW (g)	Direct Bilirubin (mg/dL)	ALP(IU/L)	M-ALP (IU/L)	Diagnosis of MBDP
Patient 1	M	28 + 1	990	3.7	366	661	No
Patient 2	F	24 + 4	760	5.2	673	806	No
Patient 3	F	26 + 1	780	5.1	357	795	No
Patient 4	M	31 + 5	1370	5.3	294	816	No
Patient 5	F	26 + 2	760	6.9	1344	971	Yes
Patient 6	F	26 + 4	1080	2.3	555	525	No
Patient 7	M	27 + 2	600	3.2	1060	612	Yes
Patient 8	M	29	810	4.7	763	756	Yes
Patient 9	M	26 + 4	740	2.4	904	536	Yes
Patient 10	F	29 + 2	915	6.2	419	903	No
Patient 11	F	31	1150	2.4	432	535	No
Patient 12	M	26 + 0	935	4.8	668	767	No
Patient 13	M	26 + 2	740	2.4	900	535	Yes
Patient 14	M	27 + 2	600	3.2	1060	612	Yes

Abbreviations: GA: Gestational Age; BW: Birth weight; ALP: Alkaline Phosphatase; M-ALP: Modified Alkaline Phosphatase; MBDP: Metabolic Bone Disease of Prematurity; M: Male; F: Female.

**Table 3 nutrients-12-03854-t003:** Number of patients from the Cholestasis Group (ChG) with and without criteria of MBDP. Number of preterm infants with serum ALP > 500 IU/L (a) and an ALP/M-ALP ratio > 1 (b).

(a)			
**ALP >500 IU/L**	**MBDP**	**No MBDP**	**Total**
Positive	6	3	9
Negative	0	5	5
Total	6	8	14
(b)			
**ALP/M-ALP >1**	**MBDP**	**No MBDP**	**Total**
Positive	6	1	7
Negative	0	7	7
Total	6	8	14

Abbreviations: ALP: Alkaline Phosphatase. M-ALP: Modified Alkaline Phosphatase. MBDP: Metabolic Bone Disease of Prematurity.

**Table 4 nutrients-12-03854-t004:** Preterm features in the Prospective Group (PG). Analytic values at 4 weeks of life, presence/absence of the DFE and PTE nucleus and MBDP condition. M-ALP was calculated only in patients with neonatal cholestasis.

Patient	Gender	GA (Weeks)	BW (g)	ALP (IU/L)	M-ALP (IU/L)	B-ALP IU/L (%)	P (mg/dL)	Ca (mg/dL)	UP (mg/dL)	U P/Cr (mg/mg)	TRP (%)	U Ca (mg/dl)	U Ca/Cr (mg/mg)	PTH (pg/mL)	Nucleus at w37 PM	Nucleus at w40 PM	MBDP Diagnosed
1	F	28 + 3	1130	745		354 (47.5)	6.9	10.5	17	0.59	95.8	29	0.09	*	None	*	No
2	F	28 + 5	995	601	573.4	222 (36.9)	4.6	8.9	14	2	73	7	0.8	28	None	None	Yes
3	M	31 + 2	2030	215		87 (40.5)	7.3	9.9	32	1.78	92	18	0.37	69	DFE + PTE	DFE + PTE	No
4	M	31 + 4	1725	312		140 (44.9)	6.6	9.5	8	0.73	97.6	11	0.75	61	None	DFE + PTE	No
5	F	25 + 4	780	307		127 (41.4)	6	10.3	22	1.83	89	12	0.73	48	DFE	DFE + PTE	No
6	F	26 + 4	885	230		88 (38.3)	5.2	9.6	9	0.9	90	10	0.6	43	None	DFE	No
7	M	26 + 3	905	763		310 (40.6)	6.5	10.1	<4			4	1.55	17	None	None	Yes
8	F	26 + 3	730	427		186 (43.6)	6.4	8.9	33	3.3	79	10	0.39	86	None	DFE	No
9	M	28	940	456		186 (40.8)	5.7	9.8	46	2.19	84	21	0.56	70	None	*	No
10	F	33 + 2	1260	333		140 (42)	7.3	10	<4			4	0.85	31	DFE	DFE + PTE	No

* Data not available at the time of the study. Abbreviations: GA: Gestational Age; BW: Birth weight; ALP: Alkaline Phosphatase; M-ALP: Modified Alkaline Phosphatase; B-ALP: Bone Specific Alkaline Phosphatase; P: Phosphate; Ca: Calcium; Cr: Creatinine; TRP: Tubular reabsorption of phosphate; U: Urine; PTH: Parathormone; PM: Postmenstrual; MBDP: Metabolic Bone Disease of Prematurity; DFE: Distal Femoral Epiphysis; PTE: Proximal Tibial Epiphysis; F: Female; M: Male.

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
