# Peer review of "Modified Serum ALP Values and Timing of Apparition of Knee Epiphyseal Ossification Centers in Preterm Infants with Cholestasis and Risk of Concomitant Metabolic Bone Disease of Prematurity"

_nutrients, 2020, doi:10.3390/nu12123854_

Round 1

Reviewer 1 Report

The paper is interesting and present some novelties in the field.

I suggest You some improvements:

  • Please add the limitations of the study (discussion section)
  • Do you think that Ultrasound assessment of knee epyphiseal ossification centers is easy to assess? Is the correct evaluations of centers by ultrasound influenced by physician experience? Please add a paragraph in this regard. Talk about similar ultrasonography assessment in the literature and explain the US experience of physicians involved in this research.
  • In materials and methods section add a paragraph that explains technical data about US studies (US equipement used, US probe with indications of frequency Hz, anterior-lateral-posterior approach?). Include how do you detect an ossification (presence of a hyperechoic body within the cartilage.... measurements?).

Author Response

Thank you very much for your comments and suggestions that improves the manuscript. As requested, we provide a point-by-point response to the reviewer’s comments

Point 1.- Please add the limitations of the study (discussion section)

Response to point 1 (in red)

We already added the limitations of the study in the discussion section that clarify the different study groups and also incuded other lilmitation suggested by reviewer number 2.

Below you can find the text included the manuscript (lines 410 to 420)

The study was conducted in a cohort of 94 preterm infants hospitalized in a University Hospital for more than four weeks but from three different consecutive periods: 2012-2015, 2016-2019 and 2020. Although there were no changes in the clinical criteria, biochemical protocol determinations or laboratories ranges, each period comprised different pretem infants thus are not strictly comparable. In the cholestatic group (ChG) were included preterm infants with this specific condition along the hospitalization period and data were strictily used to to test sensitivity and specificity of M-ALP/ALP ratio. Biochemical and clincial data from preterm infants in the retrospective group (RG) were used obtain the M-ALP formula based in ALP and DB correlation. The prospective group (PG) in which knee epishyseal ossification were tested just included 14 preterm infants. Other medical conditions as hypoglycemia or hypothyroidism as corrected are not considered in our study.

Point 2.- Do you think that Ultrasound assessment of knee epiphyseal ossification centers is easy to assess?

Response to point 2 in discussion section, lines 407 to 409

The evaluation of DFE and TPE nucleus by ultrasonography is harmless, easily reproducible and does not require a high level of expertise by performers but provides valuable information.

Point 3.- Is the correct evaluations of centers by ultrasound influenced by physician experience? Please add a paragraph in this regard.

Reponse to point 3 (also 4 and 5)

Thank you again for your inputs regarding ultrasound assessment of knee epyphiseal ossification centers. We remark that it is not influenced by physician experience (see also response to point 5). References 20 and 21 are similar ultrasonography assessment although is not oriented through BMDP.

Result section, lines 208 to 211

Regarding the presence of ossification nucleus in the same PG group, only 3 patients (30%) presented the DFE nucleus by the 37th post-menstrual week, and only 4 presented the DFE + PTE nucleus by the 40th post-menstrual week.Both inter- and intra-operator agreement for visibility of the DFE and PTE was 100%. Concordance between all ultrasound results was 100% in assessing the presence of the nucleus.

Point 4.- Talk about similar ultrasonography assessment in the literature and explain the US experience of physicians involved in this research.

Response to point 4.- 

We included the reference of Windschall 2016 (reference number 34). As in pour study, they objectives that inter-observer agreement for visibility of the knee ossification centers was 100% and inter-observer reproducibility of measurements of the ossification centers was high. We just define the visibility of DFE and PTE.

Point 5.- In materials and methods section add a paragraph that explains technical data about US studies (US equipement used, US probe with indications of frequency Hz, anterior-lateral-posterior approach?). Include how do you detect an ossification (presence of a hyperechoic body within the cartilage.... measurements?)

Response to point 5

In methods section we added a paragraph that explains technical data about US studies (lines 150 to 158)

Ultrasonographic assessment was performed using a wide-band linear transducer with ultrasound scanner Philips Affiniti 50, by two different general pediatricians separately. There were no any intra or inter-observer differences in the results. Before the study, both pediatricians were trained in the US nucleus assessment by performing ecography on healthy full-term newborns. Each patient was measured twice by both operators (general pediatricians) on two successive days. presence of an epiphyseal ossification center was defined by the presence of a measurable hyper-echoic structure with an irregular or smooth surface within the epiphyseal cartilage. When present, longitudinal and transverse images of the left knee centers were obtained with the patient in supine position and the limb extended.

Reviewer 2 Report

Despite nutritional and medical improvements MBDP remains a problem of preterm infants. The paper aims to identify a possible specific marker of disease, however some changes must be performed.

really nice idea but the paper is really difficult to understand. The amin prolem is the not good description of subjects included in the study; it is not clear if subjects analysed had hypothyroidism or hyploglicemia ecc.

Lane 156-163: there's the description of 49 patients from RG. Why in table 1 there are 77 patients? Authors must create a table with all the patients included and analyzed with their characteristics and the data available.

In every table I suggest to indicate which patient of the tables is mentioned. I don't understand why in table 2 there are 14 cholestatic patients while in table 1 are 26. In table 4 only 10 patients.

I suggest to insert tables and pictures after the text not in a separate paragraph, it will be easier to understand the data.

Paragraph 3.3. Improve the description

Figure 1. from the legend must be removed the description of the journal! 

Figure 2. Improve the description. Who is the subject of the ecographic pictures? put some pictures of no nucleus to show the differences.

Separate patients and methods description.

States the ethic committe authorization to perform the study

Author Response

Thank you very much for your comments and constructive and accurate criticism of the manuscript. As requested we provide a point-by-point response to your comments.

Despite nutritional and medical improvements MBDP remains a problem of preterm infants. The paper aims to identify a possible specific marker of disease, however some changes must be performed.

Point 1.- Really nice idea but the paper is really difficult to understand. The main problem is the not good description of subjects included in the study; it is not clear if subjects analysed had hypothyroidism or hyploglicemia ecc.

Response to point 1 (in red)

We improve the description of subjects included in each study group, with a better definition of the different groups.

As our patients are very low-weigth preterm infants thus many medical conditions can be present. We did not consider to include other medical conditions differente than cholestasis that could be associated with MBDP as we consider that will be confusing taking into accoint the purpose of the study. All these concomitant co-morbidities were treated, we did not consider that interferes with the purpose of our study. However, following your suggestions, a parragragh with the limitations of the study was included with the assesment that “other medical conditios such as hypothyroidism or hyploglicemia, although corrected and with no clincial significance in MBDP are not considrered” (lines 411 to 421)

Point 2.- Lane 156-163: there's the description of 49 patients from RG. Why in table 1 there are 77 patients? Authors must create a table with all the patients included and analyzed with their characteristics and the data available.

Response to point 2.-

We apologyzes because missunderstanding. We agree with you that it was confusing when referring to samples or patients from the different groups. Table 1 refers to samples/determinations from the retrospective group and it is already changed.

We considered that as clinical conditions are dynamic and some patients can have cholestasis and MBDP along the hospitalization period, in order to obtain the M-ALP we considered the values of DB and ALP in each determination along the hospitalization period (77 different complete determinations/profiles corresponding  to 49 preterm infants in the RG)

Point 3.- In every table I suggest to indicate which patient of the tables is mentioned. I don't understand why in table 2 there are 14 cholestatic patients while in table 1 are 26. In table 4 only 10 patients.

Response to point 3.-

You are correct and made the corresponding changes

Table 1.- Biochemical data from patients in the retrospective group. In 26 different samples/determinations cholestasis was defined by biochemical criteria.

Table 2.- Patients from the cholestasis group in whom the M-ALP and M-ALP/ALP was tested

Table 3.- Patients from the prospective group in whom the M-ALP/ALP ratio and evaluation of knee ossification centers were performed

Point 4.-  I suggest to insert tables and pictures after the text not in a separate paragraph, it will be easier to understand the data.

Response to point 4.-

Apologyzes, tables and pictures they were included were editor template asked us for. In the final manuscript they will be included after the text.

Point 5.- Paragraph 3.3. Improve the description

Response to point 5.-

Paragraph 3.3. correspond to Formatting of Mathematical Components and was asked by the editors. However in the text you can find that correponds to the regression equation of the linear correlation between the levels of ALP and DB with a correlation coefficient of 0.682 (p<0.001).

We include a brief description (lanes 297)

Point 6.- Figure 1. from the legend must be removed the description of the journal! 

Response to point 6.-

Thank you for the observation, it was already removed (lane 264)

Point 7.- Figure 2. Improve the description. Who is the subject of the ecographic pictures? put some pictures of no nucleus to show the differences.

Response to point 7.-

We improve the description including pictures of no nucleus (Lanes 266-271) Pictures E-H show the DFE cartilage without nucleus in longitudinal (E) and transverse (F) views and the PTE cartilage without nucleus in longitudinal (G) and transverse (H) views.

Point 8.- Separate patients and methods description.

Response to point 8.-

Separated, lanes 117 and 131

Point 9.- States the ethic committe authorization to perform the study

Response to point 9.-

Statement of the authorization to perform the study in methods section, lines 170-173 

Round 2

Reviewer 1 Report

I am satisfied with the reviews performed.

Author Response

Thank very much for your inputs. I really appreciated them.

Reviewer 2 Report

Thank you for your kind reply to my suggestions. Nice work.

Author Response

Thank you very much for your comments and sugestiona. I really  appreciate them.